# Establishing and Integrating a Female Genital Schistosomiasis Control Programme into the Existing Health Care System

**DOI:** 10.3390/tropicalmed7110382

**Published:** 2022-11-16

**Authors:** Takalani Girly Nemungadi, Tsakani Ernica Furumele, Mary Kay Gugerty, Amadou Garba Djirmay, Saloshni Naidoo, Eyrun Flörecke Kjetland

**Affiliations:** 1Discipline of Public Health Medicine, School of Nursing and Public Health, College of Health Sciences, University of KwaZulu-Natal, Durban 4000, South Africa; 2Communicable Disease Control Directorate, National Department of Health, Pretoria 0001, South Africa; 3Evans School of Public Policy & Governance, University of Washington, Seattle, WA 98195-3055, USA; 4Department of the Control of Neglected Tropical Diseases, World Health Organization, 1211 Geneva, Switzerland; 5Norwegian Centre for Imported and Tropical Diseases, Department of Infectious Diseases Ullevaal, Oslo University Hospital, 0424 Oslo, Norway

**Keywords:** female genital schistosomiasis (FGS), schistosomiasis, homogenous yellow patch, grainy sandy patch, health care system, South Africa

## Abstract

Female genital schistosomiasis (FGS) is a complication of *Schistosoma haematobium* infection, and imposes a health burden whose magnitude is not fully explored. It is estimated that up to 56 million women in sub-Saharan Africa have FGS, and almost 20 million more cases will occur in the next decade unless infected girls are treated. Schistosomiasis is reported throughout the year in South Africa in areas known to be endemic, but there is no control programme. We analyze five actions for both a better understanding of the burden of FGS and reducing its prevalence in Africa, namely: (1) schistosomiasis prevention by establishing a formal control programme and increasing access to treatment, (2) introducing FGS screening, (3) providing knowledge to health care workers and communities, (4) vector control, and (5) water, sanitation, and hygiene. Schistosomiasis is focal in South Africa, with most localities moderately affected (prevalence between 10% and 50%), and some pockets that are high risk (more than 50% prevalence). However, in order to progress towards elimination, the five actions are yet to be implemented in addition to the current (and only) control strategy of case-by-case treatment. The main challenge that South Africa faces is a lack of access to WHO-accredited donated medication for mass drug administration. The establishment of a formal and funded programme would address these issues and begin the implementation of the recommended actions.

## 1. Introduction

Genital grainy sandy patches and homogenous yellow sandy patches are among the gynaecological manifestations of female genital schistosomiasis, and are caused by the presence of *Schistosoma haematobium* ova in genital tissue [1]. *Schistosoma haematobium* is a blood fluke that causes disease, and if not treated, causes subsequent pathology in the urinary tract and genital tissue, the latter called female genital schistosomiasis (FGS) [1]. The global prevalence of FGS is not known, but it has been reported to be high in poor and rural communities in the tropical and subtropical parts of the world, especially those which do not have access to adequate sanitation and safe water [2,3]. It is estimated that up to 56 million women in sub-Saharan Africa have FGS, and almost 20 million more cases will occur in the next decade unless girls are treated [4]. A study in the KwaZulu-Natal province of South Africa, found one or more of the three well-known genital mucosal manifestations of FGS [5]. Of the 2008 to 2009 biopsy-diagnosed schistosomiasis cases in the Limpopo province of South Africa, 87.6% were FGS (n = 233/266) [6].

Anti-schistosomal treatment, before the development of genital lesions, forms part of the prevention of FGS [7]. Treatment before the age of 13 years offers the best protection against *S. haematobium*-induced pathology in the genital tract. However, a single dose of anti-schistosomal treatment does not seem to prevent all genital morbidity [8]. The risk of FGS in Africa is underestimated, schistosomiasis treatment is not universally available, and FGS does not appear anywhere in the health programmes [9,10,11]. Schistosomiasis is, however, reported throughout the year in focal areas, mostly where there is no continuous access to piped water. In non-endemic areas, a large number of imported cases are reported through the notifiable disease systems of South Africa [12,13]. In rural KwaZulu-Natal, South Africa, high prevalences of FGS, pregnancy, HIV, and sexually transmitted infection (STI) are reported among sexually active young women [14,15].

The burden of FGS is relative to genitourinary schistosomiasis in endemic areas [16,17]. In clinical practice, FGS is misdiagnosed as an STI or cervical cancer, resulting in unnecessary surgery and unwarranted administration of antibiotics [3,11,15,18,19,20,21,22,23]. Misdiagnosed and wrongly treated individuals therefore continue to experience FGS symptoms and have heightened risks of HIV and HPV infections [24,25]. The lack of knowledge of FGS among clinicians is exacerbated by the neglect of schistosomiasis control [17].

## 2. The Case of South Africa

### 2.1. Geography

South Africa is divided into nine provinces: the Eastern Cape, which has two metropolitan municipalities and six district municipalities; the Free State, which has one metropolitan municipality and four district municipalities; Gauteng, which has three metropolitan municipalities and two district municipalities; KwaZulu-Natal, which has one metropolitan municipality and ten district municipalities; Limpopo, which has five district municipalities; Mpumalanga, which has three district municipalities; the Northern Cape, which has five district municipalities; North West, which has four district municipalities; and Western Cape, with one metropolitan municipality and five district municipalities. The provinces vary greatly in size, with Gauteng being the smallest, most densely populated, and most urbanized. Northern Cape is the largest and least populated province. 

South Africa has three capitals: Cape Town, in the City of Cape Town metropolitan municipality, Western Cape, is the legislative capital; Bloemfontein, in Mangaung metropolitan municipality, Free State, is the judicial capital; and Pretoria, in the City of Tshwane metropolitan municipality, Gauteng, is the administrative capital, and the ultimate capital of the country.

### 2.2. Schistosomiasis Prevalence

In South Africa, both recent mappings of schistosomiasis and research on FGS among school learners from various districts and schools shed light on challenges caused by schistosomiasis requiring serious attention (Table 1).

A school survey conducted during 2016–2019 by the South African Department of Health and University of KwaZulu-Natal found that schistosomiasis prevalence was moderate (between 10% and 50%) in 32 local municipalities within five provinces, excluding Gauteng (Johannesburg area) and North West provinces, whereas one local municipality and 8 schools were identified as high risk (more than 50% prevalence) in KwaZulu-Natal (Figure 1) [34,35,36,37,38,39,40].

Although findings from the mapping revealed low prevalence in most districts, it is important to note that the mapping exercise had some limitations related to sampling criteria given the focality of the disease, timing of specimen collection, and frequency of specimen collection, e.g., collecting a single sample only. These limitations result in difficulties in making inferences on the prevalence and intensity of infection, as well as the presence of female genital lesions and urinary tract disease [13,41].

As has been found for other urban areas, Gauteng province was found to be at low-risk for schistosomiasis and reports indicate that cases may have been imported from other provinces and countries [42,43,44]. FGS screening and management should therefore be performed even in provinces that do not have local schistosomiasis transmission [11]. Without schistosomiasis mass treatment interventions and screening measures, the FGS prevalence will increase surreptitiously and continue to have obvious negative impacts such as sub-fertility or infertility, ectopic pregnancy, spontaneous abortion, premature birth, low birth weight, maternal death, genital symptoms, and increased HIV susceptibility [3,4,45,46]. If mass treatment is implemented alongside individual case management, continuous provision of piped water and community awareness, a schistosomiasis elimination target of <1% proportion of heavy intensity infections with schistosomiasis can be achieved [45]. The purpose of this policy brief is to identify action points for the control of FGS in Africa.

### 2.3. Policy Actions and Implications

The South African Department of Health has not implemented a control programme despite its policy and implementation guidelines called “Regular treatment of school-going children for soil-transmitted helminth infections and Bilharzia” [47]. Similarly, many other countries have failed to follow through with 75% treatment coverage in endemic areas and women suffer the consequences (45]. In South Africa, schistosomiasis management is limited to case-based treatment for those who seek medical care; however, genital schistosomal morbidity is seldom managed [9,47]. Some community members do not even seek medical care if they experience blood in the urine, because they believe this is normal [48]. Often, health professionals in South Africa report and take action upon discovering high-endemic sites of schistosomiasis, usually among school children, but there are no ongoing awareness programmes or systematic testing [49,50]. Moreover, the free generic treatment that is accredited by the WHO has been donated to all endemic countries but is not available to South Africans [51]. In an impassse, the South African government does not accept the WHO accreditation and the donating company does not think it should pay for the entire process of registering the donated medication in recipient countries [51].

In order to interrupt the life cycle of schistosome transmission and prevent the occurrence of new cases, many researchers recommend a combination of control measures (including a vector control strategy which has not been fully applied in most countries) [52,53,54,55,56]. This would positively contribute to controlling FGS. In brief, prevention of Female Genital Schistosomiasis and reduction of morbidity from this disease requires five main actions:Action A: Schistosomiasis prevention by establishing a formal control programme and increasing access to treatment:

Preventive chemotherapy—Worldwide, there is currently no ongoing treatment programme for FGS [57,58]. Treatment with praziquantel targets the adult worm but has no effect on the ova in the tissues; therefore, treatment should be undergone 6–8 weeks after exposure, before massive ova deposition is caused FGS [8,58]. Mass drug administration (MDA) has been implemented in some countries against schistosomiasis and is recommended by the WHO to reduce morbidity and move towards elimination [45,58]. In addition, to prevent FGS, the following should be implemented:Preventive chemotherapy in girls against FGS alongside HPV vaccination (cervical cancer prevention) in schistosomiasis endemic schools [3,59,60]. This will secure at least one round of treatment to prevent FGS and promote awareness around the genital morbidity caused by the two diseases;Hot-spot targeted administration of praziquantel and morbidity screening [59,61,62]; andUse the opportunities to treat at the endemic primary health care facilities, provide regular treatment for adults, in addition to other community members at risk of infection and school children [59].

Administration of praziquantel in other health programmes—To improve access to treatment, it is recommended that praziquantel be administered to all individuals at risk of infection in the following health programmes:During promotion of sexual and reproductive health at the reproductive health clinics in endemic areas [15,57,59];Alongside antiretroviral therapy and pre-exposure prophylaxis for HIV/AIDS [3,15]; andAlongside cervical cancer screening.
Action B: FGS Screening–FGS screening and diagnosis should be accessible at the following various platforms in the health care system and the community:

FGS diagnosis—As a common differential diagnoses, it is recommended that all individuals with symptoms of STIs and cervical cancer be screened for FGS alongside cervical cancer screening and STI testing in all health care facilities of endemic countries; cases have been reported in South Africa, Zimbabwe, Mozambique, Cameroun, Ghana, Nigeria, Egypt, Kenya, Tanzania, and Madagascar, showing that the disease is likely present on the entire African continent [1,3,15,18,58,63,64,65,66,67,68].

Index and secondary (surrounding) cases’ management—FGS is treated with praziquantel and as individualized disease management. However, hot spot intervention for community members who use the same water source as the index case should be carried out [45].Establishment of sentinel sites—At present, only crude extrapolations of the burden of FGS are possible [3]. Therefore, sentinel sites should be established in all municipalities where schistosomiasis is endemic (Figure 1 for South Africa) to establish baseline prevalence of FGS and track progress toward elimination. At these sites, FGS screening and case management should be prioritized alongside cervical cancer screening. Regular monitoring and evaluation surveys every 3 or 5 years are also appropriate for the follow-up of the control progress.

Action C: Vector Control

Vector control strategy involves freshwater molluscicides (chemical control), physical removal of snails (the intermediate host for schistosomiasis), and environmental modification [45,52]. This requires identification of possible infection sites, and identification of intermediate host snails as described by Chris Appleton and Nelson Miranda [52]. The process involves the following:Regular testing of water bodies as schistosomiasis transmission sites through the collection of snails and identification of intermediate hosts of *S. haematobium* and *S. mansoni*, followed by examination through snail dissection, crushing, and analysis, e.g., by Polymerase Chain Reaction (PCR) [52];Chemical control methods (e.g., molluscicides to kill the snails) are applied in artificial water bodies, such as irrigation channels, ditches, and farm dams, but not in large dams, natural streams, rivers, and lakes [52];Environmental control involves the removal of vegetation to remove the snails’ environment, depriving them of the sheltered niches they favour, and where appropriate, ensuring that the water flows faster than their tolerance limit of 0.3 m/s. Ideally, canals should be concrete lined and contoured, to encourage fast flow. Covering open canals near dwellings is a simple way of deterring human contact with the water [52]
Action D: Water, Sanitation, and Hygiene (WASH)–Provision of Continuous Piped Water

The provision of continuous piped water will reduce community members’ exposure to contaminated fresh water during chores such as washing and collecting water for bathing and cooking. The provision of sanitation facilities will reduce miracidial contamination of water through urine, vaginal fluids, or stool. This strategy may a require massive cost outlay but it will be cost-beneficial in the long run; it requires a long-term goal for multi-sectoral collaboration [58,64]. The strategy involves:Identification of households that do not have access to proper sanitation and piped water;Installation and maintenance of proper sanitary structures and water pipes to these households;Household education on the effective use of sanitary structures and water to improve hygiene; andContinuous health promotion campaigns and awareness on the prevention of schistosomiasis and FGS, to influence both domestic and recreational behaviour change.
Action E: Creating Awareness of Schistosomiasis and FGS

The demand for medication may be low due to a lack of awareness and lack of referrals for treatment [69]. There has never been an awareness programme for FGS [20,25,70]. This cross-cutting activity will augment all the other actions. For effectiveness and yield of the desired outcome, an awareness programme must be implemented as a parallel programme prioritizing the following key areas:Training of those who use speculums in clinical practice (clinicians, gynaecologists, nurses, and midwives) in FGS diagnosis and treatment [22,45,64]. This should include awareness of their patient’s status as an index case, active search for other cases, training in surveillance, and notification.Training of health promoters, risk communicators, environmental health practitioners, and communicable diseases control coordinators on schistosomiasis and FGS prevention and control;Training of community health workers (lay people) on schistosomiasis and FGS prevention and control;Training of water and sanitation officials in schistosomiasis and FGS prevention to assist with community awareness during installation and maintenance of sanitary structures and water pipes;Population awareness campaigns on schistosomiasis, FGS, and measures for prevention (including preventing exposure to risky freshwater) to influence behaviour change [45,66].

The relevant stakeholders for policy actions are summarized in Table 2, and are listed according to their interest and the power they have in the identified actions [71]:High power, highly interested stakeholders should be managed closely, be fully engaged, and it is important to make the greatest efforts to satisfy them;High power, less interested stakeholders are stakeholders that need to be kept satisfied, but not so much that they become bored with messages;Low power, highly interested stakeholders are stakeholders that should be kept informed with adequate information to ensure that no major issues are arising. Stakeholders in this category can often be very helpful with the details of the implementation; andLow power, less interested people are stakeholders that need to be monitored, but not bored with excessive communication.

As shown in Figure 2, the conceptual framework highlights various actions that can be implemented to decrease the burden of FGS, and reduce schistosomiasis and FGS prevalence. These actions are complementary, and can be implemented as a package.

## 3. Public Health Benefit

Health care worker knowledge, case-by-case therapy, and prevention chemotherapy with praziquantel have been proven to be effective and can improve individuals’ health in the affected communities [45,72]. In 2016, schistosomiasis caused an estimated 24,000 deaths and 2.4 million disability-adjusted life years (DALYs) worldwide [45]. One study suggested that regular anti-schistosomal MDA with praziquantel could prevent genital schistosomiasis in more than 200,000 young women in rural KwaZulu-Natal province schools in South Africa, and that by treating and preventing FGS, it would be possible to prevent more than 5000 HIV infections in adolescent girls and young women [9].

The effects would be difficult to measure as many components influence behaviour and HIV transmission, and none can be put on hold. The expected outcomes of the programme are reduced number of FGS and schistosomiasis cases, as well as reduced school absenteeism. Therefore, indirect indicators that can be used to measure public health benefit from the recommended control actions could be number of diagnosed FGS and schistosomiasis cases or unconfirmed STI symptom cases, as well as the rate of school absenteeism. “Doctor shopping” surveys for genital symptoms could be measured. A study conducted in Bandanyenje Primary School in the Manicaland Province in Zimbabwe found that annual praziquantel treatment delivered to school children over 2 years of age had a significant impact on the reduction of prevalence, intensity of infection, and reinfection of *S. haematobium* infection [73]. In another study, women who had received anti-schistosomal treatment in childhood had much less genital morbidity as adults [8]. However, such long-term effects would be difficult to measure unless FGS becomes a notifiable disease.

It is not clear whether the availability of continuous piped water will completely influence behaviour change and community members will stop exposing themselves to risky fresh water, for example, even children with taps continue to use rivers and dams for recreational purpose and laundry might be easier to complete in a river, than in small basins at home [7]. This might require vigorous community awareness and health promotion campaigns.

## 4. Ethics/Equity

The Department of Health South Africa has surveyed schistosomiasis and identified the communities at risk of infection; lack of schistosomiasis control services in these communities is unethical [34,35,36,37]. FGS has also been shown to have a significant detrimental influence on the HIV pandemic among untreated adolescent girls and young women, including those with historical unsafe water contact and who move to urban areas [9,74]. The majority of communities at risk of infection also have difficulty obtaining health services due to transportation issues or a lack of awareness about disease prevention. As a result, in order to ensure equitable distribution of resources, these populations must be prioritized for preventive chemotherapy, case management, awareness, and reproductive health care. Furthermore, the Department of Water and Sanitation is a required stakeholder for the Department of Health to campaign for piped water.

## 5. Administrative Feasibility and Budgetary Feasibility

FGS should be investigated where visual inspections are carried out using a speculum and a good light source. As a disease affecting women of reproductive age, FGS should therefore fall under the jurisdiction of cervical cancer screening, reproductive health, primary health care, and HIV prevention programmes [15]. The schistosomiasis prevention programme is within the mandates of the communicable diseases control units of the national Departments of Health, including development and implementation of control strategies. However, currently, generic medication for preventive chemotherapy is donated by Merck, via the World Health Organization, to all the affected countries. South Africa is yet to have the donated medication registered in the country. As a temporary measure, one can apply to use donated praziquantel as an exception (under Section 21 of the South African Health Product Regulatory Authority) [75]. Furthermore, South Africa has a tender for the (non-generic) schistosomiasis medication that is on the essential drug list; this is more expensive [76]. However, three pharmaceutical companies have the WHO prequalification for praziquantel (Macleods, Medopharm, and Hetero Labs) opening for competitive prices for procurement by the country [77]. HPV vaccination is carried out by the Department of Basic Education and they have already included annual school-based deworming for soil-transmitted helminths (STHs) in partnership with the Department of Health; schistosomiasis deworming could be included into this existing programme utilizing the existing resources. Stand-alone deworming programmes are expensive, and the cost per treated person is very high if many refuse [78]. Integrating FGS diagnosis and treatment into other programmes will be more cost-efficient, making it more fundable, and time-efficient.

## 6. Political Feasibility of Donated Medication in South Africa

The South African Health Product Regulatory Authority requires that all health products be approved and registered before entry and use. On the other hand, South Africa and many other countries are Member States of the World Health Assembly, and Ministers of Health have endorsed the regulations relating to the control of schistosomiasis with emphasis on mass drug administration. Ministers of Health should therefore support advocacy efforts for improved access to treatment, including entry into South Africa. In the past, TAC launched a successful campaign and put pressure on the Government of South Africa to make available AIDS treatment in public facilities [79]. TAC’s vision continues to be “engaging in monitoring, advocacy, and campaigning within the healthcare system to ensure that all public healthcare users can access quality and dignified healthcare—and that all people with HIV and TB can access prevention, treatment, care, and support services” [79]. Schistosomiasis is associated with increased risk for HIV transmission, and organizations such as Treatment Action Campaign (TAC) should be targeted for advocacy [16,63,80,81,82,83,84].

Table 3 shows the weighing score for each policy action [85]; the actions are in line with the WHO schistosomiasis control and elimination recommendations and experiences from countries such as China [45,60,86,87,88,89,90,91]. All actions were weighted as low, medium, or high for each criterion mentioned above, and based on the requirement for shorter or longer term activities for each action to impact on the expected outcome of the programme (reduced or elimination of FGS and schistosomiasis).

## 7. Implementation Challenges and Recommendations

### 7.1. Adaptive Challenges and Recommendations

The WHO-accredited Tablet Pole (measuring height) replaces the weight scale in most mass treatment campaigns [92]. In South Africa, it has been found to be inaccurate in determining the praziquantel dose due to the presence of overweight/obese children [93]. Precision can, however, be improved by adding an extra tablet of praziquantel to the standard dose.

In most countries, medication is dispensed by teachers as recommended by the WHO [92]. In South Africa, however, praziquantel is a “schedule 4” drug and must be dispensed by nurses or doctors. Teachers are currently dispensing mebendazole for soil-transmitted helminths. However, they have not taken on anti-schistosomal medication, because it is “illegal” and furthermore, it could be seen as a significant burden and additional obligation for teachers to use a weight scale or the WHO Tablet Pole if not properly explained [93]. In South Africa, therefore, school health nurses dispense praziquantel, greatly increasing the cost of FGS prevention [9,78,94]. Regulatory issues must be addressed and subsequently, teachers, their unions, and school governing bodies should be addressed [92].

Finally, girls could, alongside HPV vaccination/deworming to prevent cervical cancer/STHs, also be given praziquantel, to prevent FGS.

### 7.2. Regulatory Challenges for Praziquantel, Specifically for South Africa and Recommendations

In South Africa, praziquantel is dispensed by registered nurses or doctors even though it is very safe and has been dispensed by teachers for four decades in Asia, South America, and the rest of Africa. The South African regulation drives the cost up and highly impractical in mass treatment scenarios. Furthermore, studies have found that there is better compliance if teachers are dispensing it [92]. Additionally, two brands of praziquantel are registered in South Africa but not the WHO-donated drug. As a result, the cost of mass treatment is much higher in South Africa than in similar countries. Therefore, the following is recommended:Praziquantel should be down-regulated from schedule 4 to schedule 1.The Department of Health must apply to the South African Health Products Regulatory Authority under Section 21 of the South African Health Product Regulatory Authority for the admission of donated pharmaceuticals.The disease is neglected and will require dedicated coordinators. Staff turnover may have an influence on program implementation. Therefore, integration into the existing programmes and improving awareness among health care workers are essential.FGS should be incorporated in the “Regulations relating to the surveillance and the control of notifiable medical conditions” [12].Registration of the WHO donated praziquantel, Cesol^®^, Merck, Mexico, should be prioritized instead of using Section 21 of the South African Health Product Regulatory Authority exemption on an annual basis. This can be achieved through strong advocacy and evidence-based motivation.Ensure continued availability of praziquantel in all known endemic primary health care facilities, mother and child clinics, cervical cancer screening sites, HPV vaccination programmes, PrEP, and other HIV prevention programmes.

## 8. Technical Issues and Recommendations

Colposcopes are generally not available in low-resource settings for visualization of the lesions, and the training of clinicians and other health care workers on schistosomiasis and FGS diagnostic skills should be carried out [17,20]. Health information officers, data collectors, and personnel must be trained in order to increase knowledge and reporting/notification and data collection tool must be developed.

## 9. Laboratory Analysis for FGS

Clinical (visual) recognition is essential for FGS diagnosis and cannot be replaced by urine microscopy. However, PCR of genital specimens may support the FGS diagnosis. The Circulating Anodic Antigen (CAA) test (Leiden University Medical Centre, The Netherland and Ampath, South Africa) may be used to monitor progress towards elimination at the sentinel sites and may be useful for screening as an alternative to urinary microscopy, especially where prevalence and intensity of infection are low. The CAA has been found to be a highly sensitive and 100% specific test, capable to detect extremely low concentrations of the parasite-excreted CAA, potentially down to the level of a single worm infection [95,96].

Actions A (FSG diagnosis), B (improve access to treatment), and D (water, sanitation and hygiene) are recommended as immediate priorities since they will lead to the decrease in FGS burden, as well as the decrease or elimination in schistosomiasis and FGS. Actions C (vector control) and E (creating awareness) are recommended as long-term measures to support sustainability of the control programme and eventual elimination.

## 10. Conclusions

Africa lacks an FGS control programme and there is lack of knowledge among health care workers and communities. Integration of FGS diagnosis into the existing health care system presents opportunities to aid effective and efficient implementation of the programme [10,11].

Clinical misdiagnosis will result in untreated FGS in women and girls with symptoms of unknown cause [69]. FGS patients should be viewed as index cases with associated infections. It has been shown that local transmission may occur in poor urban settings and it is important to determine the source of urban infections and manage them [42,97,98,99,100,101,102].

In order to move towards elimination, a 2030 WHO goal, treatment programmes, and mass drug administration should be carried out at schools, in reproductive health facilities, and in the impacted communities [45,58]. To increase access to treatment in South Africa through the acceptance of WHO accredited donated medicine, resolution is likely to require a political stance or a court case. Improved access to schistosomiasis treatment will decrease the risk of new FGS and enhance the health in the affected communities. To realize the wish for treatment, community awareness must run concurrently with training of health care professionals as well as community health care workers. Although vector control, water, sanitation, and hygiene initiatives are expensive and will only be effectuated in the long term, they are vital; community awareness will play a large role by offering knowledge on actions for reducing unsafe water exposure.

## Figures and Tables

**Figure 1 tropicalmed-07-00382-f001:**
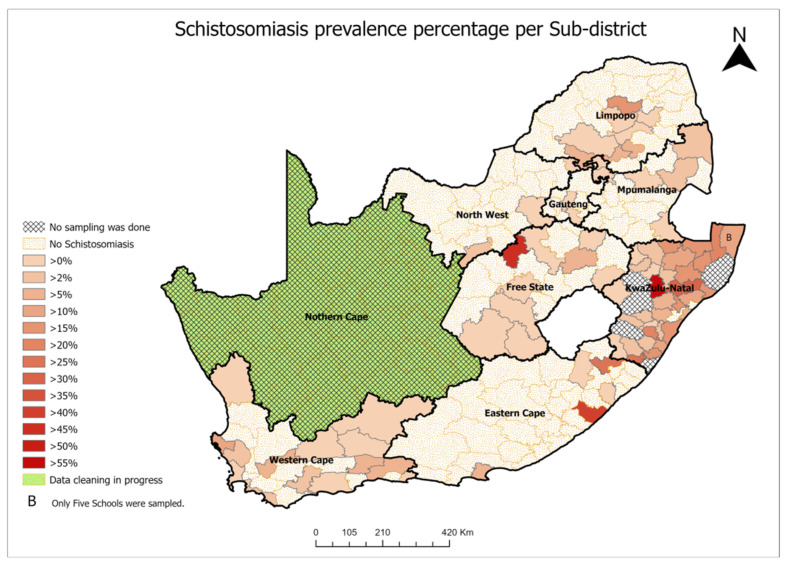
Distribution of schistosomiasis in South Africa.

**Figure 2 tropicalmed-07-00382-f002:**
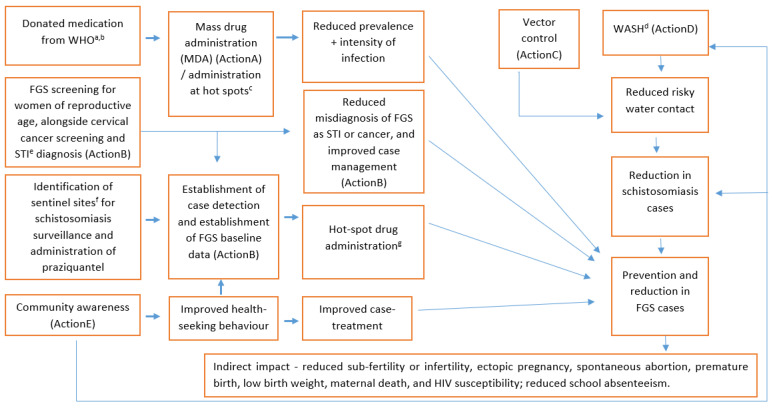
Conceptual framework to guide identification of actions for control of female genital schistosomiasis (FGS). ^a^ World Health Organization, ^b^ To our knowledge, approval is still required in South Africa to use the donated medication from WHO, ^c^ Places where clusters of cases are detected, ^d^ Water, sanitation and hygiene, ^e^ Sexually transmitted infection, ^f^ Health care facilities identified as sites for monitoring of disease pattern and progress of the control programme, ^g^ Administration of praziquantel at the emerging hot spots where clusters of cases are detected.

**Table 1 tropicalmed-07-00382-t001:** Schistosomiasis prevalence studies in South Africa.

Year of Study	Study Area	Province	Number of Schools (Total Number of Participants)	Prevalence of *S. haematobium*	Prevalence of *S. mansoni*	Mean Intensity of *S. haematobium* Infection (Eggs/ 10 mL Urine)
2009	Ugu District [9]	KwaZulu-Natal	5 (Not reported (NR))	≥50%	NR	NR
2009–2010	Ugu District [26]	KwaZulu-Natal	15 (726) ^a^	36.9%	0%	20 (range from 1–624)
2009–2010	Ugu District [7]	KwaZulu-Natal	18 (970)	32%	0%	52
2010	Ugu District [27]	KwaZulu-Natal	18 (1057)	32%	0%	52
2010 2011	Ugu District ^b,f^ [13]	KwaZulu-Natal	18 (108) 18 (76)	38% and 29.6% ^c^ 13.2% ^f^ and 11.8% ^c^	NR	18.4 ^b^ and 14.9 ^c ^ 9 ^b^ and 12.9 ^c^
2011	Ugu District [9]	KwaZulu-Natal	9 (NR)–	10–49%	NR	NR
2011	Ugu District [9]	KwaZulu-Natal	2 (NR)	<10%	NR	NR
2011	Ugu, iLembe, and Southern uThungulu (King Cetshwayo) districts [9]	KwaZulu-Natal	2 (NR)	≥50%	NR	NR
2011	Ugu, iLembe, and Southern uThungulu (King Cetshwayo) districts [9]	KwaZulu-Natal	47 (NR)	10–49%	NR	NR
2011	Ugu, iLembe, and Southern uThungulu (King Cetshwayo) districts [9]	KwaZulu-Natal	13 (NR)	<10%	NR	NR
2011–2013	iLlembe, Ugu and uThungulu (King Cetshwayo) districts [5]	KwaZulu-Natal	NR (1123)	26.0%	NR	
2012	Ugu District [28]	KwaZulu-Natal	3 (246) and 9 ^d^ (873)	20.4%	NR	14
2014	uMkhanyakude (Jozini Municipality) [29]	KwaZulu-Natal	10 (420)	40.2%	NR	NR
2015	uMkhanyakude District (Ndumo) [30]	KwaZulu-Natal	10 (320)	37.5%	NR	NR
2019	uMkhanyakude District [31]	KwaZulu-Natal	34 (1143) ^e^	1.0%	0.9%	30.4
2004 2005 2005 2005 2005 2005	Vhembe District [32]	Limpopo	NR	>70% ^f^	NR	NR
2005	Vhembe District [32]	Limpopo	1 (94)	36.2% ^g^	NR	NR
2005	Vhembe District [32]	Limpopo	NR (148)	42% ^h^	NR	NR
2005	Vhembe District [32]	Limpopo	1 (247)	86% ^e^	NR	NR
2005	Vhembe District [32]	Limpopo	1 (191)	84% ^e^	NR	NR
2005	Vhembe District [32]	Limpopo	1 (138)	78.2% ^e^	NR	NR
2009–2013	Rob Ferreira Hospital Patients [33]	Mpumalanga	304 ^i^	10.2%	NR	NR

NR = not reported. Population in rows 1,4, and 6–9 may partially overlap. ^a^ Public primary schools. ^b^ Measured during hot season. ^c^ Measured during cold season. ^d^ High schools. ^e^ Preschools and early childhood development (ECD) centres. ^f^ Retrospective laboratory data analysis for the year 2004 in major hospitals in Vhembe District (indicated as hospital A, B and C in the published paper) among patients who attended the main hospitals with urinary tract infection. ^g^ University of Venda students. ^h^ Primary school learners. ^i^ Appendix samples removed in theatre, Rob Ferreira Hospital for histological investigation (microscopy for schistosomiasis). Results were not classified according by schistosome species.

**Table 2 tropicalmed-07-00382-t002:** Stakeholders for each policy action.

Policy Actions	Strategic Activities	Stakeholder at the Forefront with Both High Power and High Interest ^a^	Other Stakeholders and Their Interest
Action A: FGS screening	FGS diagnosis Case management Establishment of sentinel sites for surveillance	Department of Health ^a^ Gynaecological services, cervical cancer screening programme, STI clinics ^a^ Department of Vector control and Public Health, National Institute for Communicable Diseases (NICD)	Department of Education and School Governing Body have high interest ^a^ and low power Community members including leaders, learners and their parents have high interest ^a^ and low power Research community has high interest ^a^ and low power The World Health Organization (WHO) has high interest and in-country low power
Action B: Treatment for Schistosomiasis	Praziquantel (Mass drug administration/administration at hot spots/Case treatment)	Department of Health ^a^ Department of Education ^a^ WHO	School Governing Body ^b^ have high interest and low power Community members including leaders, learners and their parents have high interest ^a^ and low power South African Health Products Regulatory Authority have high power but low interest Research community has high interest and low power
Action C: Vector Control	Freshwater snail control with molluscicides (chemical control) Physical removal of snails, and environmental modification	Department of Health Department of Environmental Affairs Department of Vector control, NICD	Department of Agriculture, Land Reform and Rural Development has high power and low interest Research and Academic Institutions have high interest and less power WHO has high interest and low power
Action D: Water, Sanitation and Hygiene (WASH)	Provision of continuous piped water Provision of proper sanitary facilities Hygiene education Training and awareness campaigns on schistosomiasis and FGS	Department of Water and Sanitation Department of Health Municipalities and South African Local Government Authority	Community members including leaders, learners and their parents have high interest ^a^ and low power WHO has high interest and low power Research and Academic Institutions have high interest and low power
Action E: Creating Awareness of Schistosomiasis and FGS	Training and awareness on schistosomiasis and FGS among health care professionals Awareness campaigns in the community on schistosomiasis and FGS	Department of Health WHO Department of Education Department of Water and Sanitation Municipalities and South African Local Government Authority National Health Laboratory Service (control programme and diagnostics, research, profit and improved public health) Research community and Academic Institution has interest in the control programme and improved public health School Governing Body	Department of Agriculture, Land Reform and Rural Development has less interest and low power Department of Environmental Affairs has low interest and low power

Interest requires knowledge and awareness. Currently most health professionals do not know about FGS [11,70]. ^a^ When they are knowledgeable and aware of FGS, ^b^ Elected by parents.

**Table 3 tropicalmed-07-00382-t003:** Policy actions matrix for Africa.

Action	Criteria			
	Public Health Impact (Efficacy)	Ethics/Equity	Administrative Feasibility	Budgetary Feasibility
A. Prevention and increased access to treatment	High	High	High	High
B. FGS diagnosis	High	High	High	High
C. Vector control	High	Medium	Medium	Medium
D. Water, sanitation and hygiene (WASH)	High	High	Low	Low
E. Creating awareness	High	Medium	High	High

Low weight means poor, medium means moderate, and high weight means good.

## Data Availability

Data sharing is not applicable; no new data created.

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
