# Peer review of "Establishing and Integrating a Female Genital Schistosomiasis Control Programme into the Existing Health Care System"

_tropicalmed, 2022, doi:10.3390/tropicalmed7110382_

Round 1
Reviewer 1 Report
This is a well researched paper, informative, and a valuable resource for addressing the schistosomiasis problem in Africa, in particular, female genital schistosomiasis (FGS)
Author Response
Review report is noted. Thank you.
Reviewer 2 Report
Thank you for giving me the opportunity to review the article. The authors conducted a review focusing on the female genital schistosomiasis control program in South Africa. I thought that the topic is socially important, but there are several problems in the manuscript. Therefore, the authors should revise the manuscript before further considerations. I listed my comments below.
Comments:
The case of South Africa:
1. What does "Row" in the Table 1 mean?
2. The several cells for “Year of study” are blank.
3. Why did the authors mention about the number of schools in the Table 1? The authors should explain the context in the main text.
4. The authors should add the reference number (referenced article number) for each study in the Tabel 1.
5. It is difficult to understand the urban and rural area from the map (Figure 1) and the main text. Therefore, the authors should explain the geographical situation of South Africa.
6. The authors should explain about the geographical differences of the female genital schistosomiasis control program in more detail.
7. The author should check the styles of the tables. They should make them consistently.
8. How did the authors separate “stakeholders” and “other stakeholders” in the Table 2? They should describe about it.
Public health benefit:
9. The authors should discuss about to measure public health benefit in the future.
Administrative feasibility and budgetary feasibility:
10. The authors should describe about the costs and the budgets in more detail (with the data).
Political feasibility of donated medication in South Africa:
11. How did the authors categorize (high/moderate/low) the policy actions.
Technical issues and recommendations:
12. Can the authors refer the solutions which used in other countries with appropriate references?
References:
13. The authors should check and correct the styles of references.
Ref: https://www.mdpi.com/journal/tropicalmed/instructions
Author Response
Thank you. Please see attachment.

Reviewer 3 Report
Dear Respect Editor
The manuscript entitled: “establishing and integrating a female genital schistosomiasis control programme into the existing health care system, the case study of South Africa” addresses a highly remarkable topic. This is an interesting paper which serves to highlight the female genital schistosomiasis control in Africa. Schistosomiasis is one of the important parasitic diseases that impose severe complications on human society. As one of at risk the groups for this disease, pregnant women has double importance. Considering the importance of the disease, I suggest the publication of this nice manuscript.
Author Response
Review report is noted. Thank you
Reviewer 4 Report
Please find the report in an attachment

Reviewer 5 Report
Thank you for sharing this article on setting up an FGS control program in schistosomiasis endemic settings of South Africa. Overall, I would recommend to customise the proposed action points more to the context of South Africa. The way the action points under 2.2 are phrased they appear to be applicable universally in schistosomiasis endemic settings. Table 2 has some aspects specific to South Africa, but again Figure 2 appears rather general and not much related to afflicted settings in South African. Likewise, the policy action matrix shown in Table 3 seems rather general as already its title states "policy actions matrix for Africa and expected outcome". Here some minor comments that could help to improve the article.
L41: Not clear what you mean by "cases 2008 to 2009". Do you mean cases recruited during 2008 to 2009?
L47: Not clear what you man by "schistosomiasis treatment not universally available". Consider rephrasing "schistosomiasis treatment is not universally available".
L49/50: Consider rephrasing to "where there is no continuous access to piped water".
L51-53: Consider rephrasing "In rural Kwa-Zulu-Natal, South Africa, there are reports of a high prevalence of FGS, pregnancy, HIV, STIs among sexually ative young women" to "In rural Kwa-Zulu-Natal, South Africa, high prevalence of FGS, pregnancy, HIV, and STIs are reported among sexually active young women."
L56: Change "unwarranted antibiotics" to "unwarranted administration of antibiotics".
L58: Change "HIV and HPV" to "HIV and HPV infections".
L64: Consider rephrasing "light on schistosomiasis as a challenge requiring serious attention" to "light on challenges caused by schistosomiasis requiring serious attention".
Table 1: Rephrase "prevalence for S. haematobium" and "prevalence for S. mansion" to "prevalence of S. haematobium" and "prevalence of S. mansion". What is the meaning of "row" in the table?
L71: Change "a 2016-2019 survey of schools conducted" to "a school survey conducted during 2016-2019".
L79: Rewrite "although mapping findings" to "although findings from mapping".
L81: Change "specimen collection timing" to "timing of specimen collection"
L81-82: Reword "only one specimen was collected" to "frequency of specimen collection, e.g., collecting a single sample only".
L82: Correct "results" to "result".
L83: Change "presence of genital lesions and urinary tract disease in females" to "presence of female genital lesions and urinary tract disease".
L94: Rephrase "provision of continuous piped water" to "continuous provision of piped water".
L95: Is the target of <1% heavy infection addressed at S. haematobium only or schistosomiasis in general disregarding of the species?
L96: Consider changing " actions" to "action points".
L106: Delete "sometimes".
L108: Regarding "a high-endemic sites", rephrase to "high-endemic sites" or "a high-endemic site".
L107-109: Not clear what the sentence "sporadically ... testing" means; please rewrite.
Author Response
L41: the issue is addressed. These were the cases diagnosed and recorded during 2008 to 2009.
L47: the issue is addressed in the manuscript
L49/50: the issue is addressed in the manuscript
L51-53: the issue is addressed in the manuscript
L56: the issue is addressed in the manuscript
L58: the issue is addressed in the manuscript
L64: the issue is addressed in the manuscript
L71: the issue is addressed in the manuscript
L79: the issue is addressed in the manuscript
L81: the issue is addressed in the manuscript
L81-82: the issue is addressed in the manuscript
L82: the issue is addressed in the manuscript
L83: the issue is addressed in the manuscript
L94: the issue is addressed in the manuscript
L95: the issue is addressed in the manuscript
L96: the issue is addressed in the manuscript
L106: the issue is addressed in the manuscript
L108: the issue is addressed in the manuscript
L107-109: the issue is addressed in the manuscript
Round 2
Reviewer 5 Report
The authors have addressed all my previously recommended changes sufficiently.